# Serum Levels of Soluble Triggering Receptor Expressed on Myeloid Cells-1 Associated with the Severity and Outcome of Acute Ischemic Stroke

**DOI:** 10.3390/jcm10010061

**Published:** 2020-12-26

**Authors:** Jyun-Bin Huang, Nai-Ching Chen, Chien-Liang Chen, Mu-Hui Fu, Hsiu-Yung Pan, Chung-Yao Hsu, Shang-Der Chen, Yao-Chung Chuang

**Affiliations:** 1Department of Emergency Medicine, Kaohsiung Chang Gung Memorial Hospital, Kaohsiung 83301, Taiwan; u9001135@gmail.com (J.-B.H.); fornever@cgmh.org.tw (H.-Y.P.); 2College of Medicine, Chang Gung University, Taoyuan 33302, Taiwan; naiging@yahoo.com.tw (N.-C.C.); k8601085@cgmh.org.tw (M.-H.F.); chensd@adm.cgmh.org.tw (S.-D.C.); 3Department of Neurology, Kaohsiung Chang Gung Memorial Hospital, Kaohsiung 83301, Taiwan; 4Division of Nephrology, Kaohsiung Veterans General Hospital, Kaohsiung 813, Taiwan; cclchen@seed.net.tw; 5Department of Medicine, National Yang-Ming University School of Medicine, Taipei 112, Taiwan; 6Department of Neurology, School of Medicine, College of Medicine, Kaohsiung Medical University Hospital, Kaohsiung Medical University, Kaohsiung 80708, Taiwan; cyhsu61@gmail.com; 7Institute for Translation Research in Biomedicine, Kaohsiung Chang Gung Memorial Hospital, Kaohsiung 83301, Taiwan; 8Department of Biological Science, National Sun Yat-sen University, Kaohsiung 80424, Taiwan

**Keywords:** soluble triggering receptor expressed on myeloid cells-1, acute ischemic stroke, neuroinflammation, National Institutes of Health Stroke Scale, stroke severity, functional outcome

## Abstract

Stroke is a neurological emergency, where the mechanism of the blood supply to the brain is impaired, resulting in brain cell ischemia and death. Neuroinflammation is a key component in the ischemic cascade that results in cell damage and death after cerebral ischemia. The triggering receptor expressed on myeloid cells-1 (TREM-1) modulates neuroinflammation after acute ischemic stroke. In the present study, 60 patients with acute ischemic stroke, who had been subjected to neurological examinations and National Institutes of Health Stroke Scale (NIHSS) and brain magnetic resonance imaging studies, were enrolled in the emergency room of Kaohsiung Chang Gung Memorial Hospital. Twenty-four healthy volunteers were recruited as controls. The serum levels of soluble TREM-1 (sTREM-1), human S100 calcium-binding protein B (S100B), and proinflammatory cytokines and chemokines, including tumor necrosis α (TNF-α), interleukin 1β, interleukin 6 (IL-6), interleukin 8, and interferon-γ were measured immediately after acute ischemic stroke. The serum levels of sTREM-1, TNFα, IL-6, and S100B were correlated with the stroke volume and NIHSS, after acute ischemic stroke. Additionally, the serum levels of sTREM-1 were significantly positively correlated with S100B. The functional outcomes were evaluated 6 months after ischemic stroke by the Barthel index, which was correlated with the age and levels of sTREM-1 and S100B. We suggest that acute ischemic stroke induces neuroinflammation by the activation of the TREM-1 signaling pathway and the downstream inflammatory machinery that modulates the inflammatory response and ischemic neuronal cell death. From a translational perspective, our results may allow for the development of a new therapeutic strategy for acute ischemic stroke by targeting the TREM-1 signaling pathway.

## 1. Introduction

Stroke is a medical and neurological emergency, where the mechanism of the blood supply to the brain is impaired, resulting in brain cell ischemia and death [1,2]. Among the great number of human diseases, stroke is the third leading cause of death, after heart disease and cancer. It leads to permanent disabilities in 80% of survivors [3,4]. About 70–80% of stroke patients are ischemic, and the remaining patients are hemorrhagic [1,2]. Early treatment of acute ischemic stroke is crucial. It can minimize brain damage and complications and improve outcomes. Thrombolysis with recombinant tissue plasminogen activator (rtPA) and endovascular thrombectomy are the main revascularization therapies for acute ischemic stroke [5,6,7,8,9]. However, even if the blockage associated with acute ischemic stroke is removed early, ischemia-reperfusion injury after revascularization therapy can aggravate the neurological complications and result in worse outcomes [10,11,12]

In addition to ischemia-reperfusion injury, following ischemic stroke, post-stroke secondary neuroinflammation is responsible for critical pathological processes that promote further cell injury, resulting in ischemic brain cell death, disability, and poor outcomes [11,13,14,15]. The post-stroke immune response consists of the aberrant activation of glial cells, particularly microglial cells, infiltration of peripheral leukocytes, and release of damage-associated molecular pattern (DAMP) molecules from the damaged cells of ischemic brain [11,16]. DAMPs can stimulate immune cells to produce proinflammatory cytokines and chemokines, adhesion molecules, and several immune molecule effectors and augment oxidative/nitrosative stress, which leads to the exacerbation of cerebral ischemic injury [11,13,16,17]. High levels of proinflammatory cytokines and chemokines released early, after the onset of brain ischemia, may be associated with stroke severity and a worse prognosis [13,18,19].

The triggering receptor expressed on myeloid cells-1 (TREM-1), which belongs to the immunoglobulin family of cell surface receptors, is a recently identified molecule involved in myeloid cell activation and innate immunity [20,21]. TREM-1 is an orphan receptor, and it has been reported to be associated with toll-like receptor 4 (TLR4), as a critical amplifier of inflammatory signaling in various diseases [21,22]. Thus, evidence has shown that the TREM-1 pathway may be implicated as a therapeutic target during acute or chronic inflammatory or infectious conditions and cancer [23,24,25]. Moreover, circulatory concentrations of soluble TREM-1 (sTREM-1) have been suggested as a clinically valuable diagnostic and prognostic biomarker in patients with infectious and inflammatory diseases [24,25,26]. In contrast to the innate immunity against infections, recent evidence has shown that the TREM-1 receptor and its signaling pathways contribute to the pathology of several non-infectious and non-inflammatory neurological diseases, such as cerebrovascular atherosclerosis [27,28], ischemia stroke [29,30,31], subarachnoid hemorrhage [32,33], Alzheimer’s disease [34,35,36], and Parkinson’s disease [34]. During acute ischemic stroke, endogenous molecules from damaged ischemic brain tissues are presumed to activate the TREM-1 signaling pathway and the downstream inflammatory machinery [29,30,31]. However, whether the augmentation of neuroinflammatory markers in acute ischemic brain tissue by the activation of the TREM-1 signaling pathway, after acute ischemic stroke, is associated with stroke severity and predisposes patients to neurological deterioration is unclear.

The present study evaluated the hypothesis that the serum levels of sTREM-1 and downstream neuroinflammatory pathway may play a crucial role in the severity of acute brain damage and functional outcomes, following acute ischemic stroke. Therefore, the following parameters were immediately measured in patients after acute ischemic stroke in an emergency room: the serum concentrations of endogenous sTREM-1, proinflammatory cytokines and chemokines, and human S100 calcium-binding protein B (S100B); a marker for blood-brain-barrier (BBB) disruption and acute damage to the brain parenchyma. The relationships between these circulatory biomarkers and the severity and outcomes of ischemic stroke-survivors were also assessed.

## 2. Experimental Section

### 2.1. Patients and Study Design

This was a single-center, prospective, and observational study conducted at Kaohsiung Chang Gung Memorial Hospital. Kaohsiung Chang Gung Memorial Hospital is a tertiary medical center in Taiwan. Ethical approval for this study was provided by the Chang Gung Medical Foundation Institutional Review Board (approved number: 201801558B0), and written informed consent was obtained from the patients or their family, as well as from the controls. All procedures performed in this study were in accordance with the 1964 Helsinki declaration.

From October 2018 to October 2019, 60 patients with first-ever ischemic stroke, who had been subjected to neurological examinations and National Institutes of Health Stroke Scale (NIHSS) [37] and brain computed tomography studies in the emergency room of Kaohsiung Chang Gung Memorial Hospital, were enrolled in this study. Further brain magnetic resonance imaging studies were performed to confirm new-onset acute ischemic stroke within 48 h of admission. Concerning the delay of the treatment of acute ischemic stroke, we excluded the patients who had received intravenous rtPA thrombolysis or intra-arterial thrombolysis and mechanical thrombectomy therapies [38]. Other exclusion criteria (also see Appendix A) from the study included transient ischemic attack, brain injury, concurrent renal or hepatic insufficiency, malignancy, hematological diseases, immunological or inflammatory diseases, immunomodulatory therapy, recent surgery or trauma, and recent infection. Infection was defined as leukocytosis from blood samples in the emergency room or clinical symptoms of an infection (fever and/or pyuria for urinary tract infection and fever and/or productive cough and radiographic evidence of the consolidation of pneumonia). Twenty-four healthy volunteers, who received an annual physical checkup, were recruited as controls.

The risk factors and clinical and demographic data of the patients were collected. The subtype of acute ischemic stroke in patients was classified according to the Trial of Org 10,172 in Acute Stroke Treatment (TOAST) [39] for the classification of large or small infarctions. Patients who were found to have cerebral cortical impairment, brain stem or cerebellar dysfunction, or brain imaging findings of either significant (>50%) stenosis or the occlusion of a major brain artery or branch cortical artery, presumably due to atherosclerosis, were classified as large-artery atherosclerosis. Cardioembolism included patients with arterial occlusions, presumably due to an embolus arising in the heart. Small-artery occlusion (lacune) was defined as infarction over the territory of deep perforating arteries, including the brain stem, basal ganglia (caudate nucleus, lentiform nucleus, internal capsule, or thalamus), and white matter (corona radiata or centrum semiovale) [39,40]. Furthermore, stroke severity was determined by the NIHSS [37]. The patients were divided into three groups. Group 1 consisted of patients with an NIHSS ≤ 5, Group 2 consisted of those patients with an NIHSS between 6 and 16, and Group 3 consisted of those with an NIHSS ≥ 17. The Barthel index was evaluated 6 months after the stroke to evaluate the functional outcome of patients.

### 2.2. Assessment of Blood Leukocyte and Serum Levels of Soluble Triggering Receptor Expressed on Myeloid Cells-1 and Human S100 Calcium-Binding Protein B

Venous blood samples for biochemical study were promptly obtained in the emergency room. The leukocyte (white blood cell, WBC) count was rapidly analyzed by the central laboratory of Kaohsiung Chang Gung Memorial Hospital. The serum samples were separated by centrifugation (2000× *g* for 10 min at 4 °C) and were immediately frozen at −80 °C for analysis of the circulating biomarkers. The serum level of sTREM-1 and human S100B were measured by enzyme-linked immuno-sorbent assay kits for human TREM-1 (ELISA-20190102, R&D Systems, Minneapolis, MN, USA) and human S100B (EZHS100B-33K, Millipore, Burlington, MA, USA). All conditional medium samples were incubated in primary antibody-coated wells and further incubated with secondary antibody-horseradish peroxidase (HRP) labeled streptavidin. Antigen–antibody complexes were detected by tetramethylbenzidine substrate (NeA-Blue, Clinical Science Products Inc., Mansfield, MA, USA), as a substrate solution. The OD450 nm values were read using a Bio-Rad Benchmark Plus Multiplate Spectrophotometer (Bio-Rad, Hercules, CA, USA), and the concentrations were determined by its software. All samples were in duplicate, and the average of the two was used for statistical analysis.

### 2.3. Cytokine Multiplex Bead Immunoassay

The serum levels of proinflammatory cytokines, including tumor necrosis α (TNF-α), interleukin 1β (IL-1β) and interleukin 6 (IL-6), and chemokines, including interleukin 8 (IL-8) and interferon-γ (IFN-γ), were detected by immunology multiplex assay using a MILLIPLEX MAP Human Cytokine/Chemokine Magnetic Bead Panel (Millipore), according to the manufacturer’s protocol. The samples were centrifuged for 12,000× *g* for 10 min to remove debris, before being applied to a 96-well plate for incubation, with beads coated with antibody, which was labeled in fluorescent. Each serum and standard were triplicated, and the average value was used for analysis. The plates were read using the Luminex FlexMap 3D system (Luminex). For the quantitative analysis of the samples, a Five Parameter Logistic (5PL) curve fit was employed to calculate the concentrations from the software provided by the Luminex FlexMap 3D system.

### 2.4. Statistical Analyses

All values are expressed as the mean ± standard deviation (SD). The normality of the distribution was tested by the Kolmogorov–Smirnov test. Non-parametric methods were used when the continuous variables had a non-normal distribution. Mann–Whitney U tests were applied to compare the difference between the two groups, and Bonferroni correction was used. The Kruskal–Wallis test was used to compare continuous variables between groups. Spearman rank correlation was used to determine the relationships between NIHSS and serum biomarkers, including sTREM-1, human S100B pro-inflammatory cytokines (TNF-α, IL-1β, and IL-6), and chemokines (IFN-γ and IL-8), in 60 patients with acute ischemic stroke. Spearman rank correlation was also used to determine relationships between the Barthel index and age, NIHSS, sTREM-1, IL-6, TNFα, and human S100B. In addition, we used the Kolmogorov-Smirnov method to check the Barthel index for normal distribution, finding that *p* < 0.001 (non-normal distribution). We defined the group with a Barthel index score < 60 as the poor outcome group, and the group with a score ≥ 60 as the better outcome group [41,42]. Therefore, the factors that significantly correlated with the Barthel index were subjected to binary logistic regression analysis to determine the factors associated with functional outcomes in patients with ischemic stroke. The statistical analysis was performed using the IBM SPSS Statistics Version 22.0 (IBM, Armonk, NY, USA). A *p* value < 0.05 was considered statistically significant.

## 3. Results

### 3.1. Demographic Data of 60 Patients with Acute Ischemic Stroke

The demographic data and underling diseases of all patients are listed in Table 1. There were 60 patients with confirmed acute ischemic stroke, including 20 females and 40 males, with a mean age of 67.23 ± 10.34 (mean ± standard deviation) years, ranging from 45–86 years. The 24 normal controls included 7 females and 17 males aged 66.75 ± 9.53 years. With respect to gender and age, there were no statistically significant differences between the patients and control subjects.

To evaluate the subtypes of acute ischemic stroke, 27 patients were classified as large-infarction patients, including 16 with large-artery atherosclerosis and 11 with cardioembolism, and 33 patients were classified as small-infarction patients, according to the TOAST classification. Based on the analysis, there were no significant differences between the small-infarction and large-infarction groups in terms of gender and age.

### 3.2. Serum Levels of the Soluble Triggering Receptor Expressed on Myeloid Cells-1, Proinflammatory Cytokines, and Human S100 Calcium-Binding Protein B Correlated with the Infarct Size of Acute Ischemic Stroke

The results of the statistical analysis show that the serum levels of sTREM-1 and S100B were significantly higher in the large-infarction group than in the control group and small-infarction group (Table 2). Furthermore, the serum levels of proinflammatory cytokines, including IL-6 and TNF-α, were significantly higher in the large-infarction group compared to the control group. The serum levels of IL-6 and TNF-α were not significantly increased in patients with acute small infraction. Moreover, the difference between the small and large infarction groups and the control group in terms of the WBC count, IL-1β, IL-8, and IFN-γ was not statistically significant (Table 2).

### 3.3. Serum Levels of the Soluble Triggering Receptor Expressed on Myeloid Cells-1, Proinflammatory Cytokines, and Human S100 Calcium-Binding Protein B Correlated with the Severity of Acute Ischemic Stroke

According to NIHSS, 25 patients were assigned to Group 1 (NIHSS ≤ 5), 18 patients to Group 2 (NIHSS: 6–16), and 17 patients to Group 3 (NIHSS ≥ 17). The mean age of Group 1, Group 2, and Group 3 was 65.60 ± 10.32, 66.0 ± 10.05, and 70.94 ±10.34 years, respectively. There was no significant difference in terms of age and gender between the patients and controls (*p* = 0.356 and 0.834, respectively). Table 3 lists the correlation analysis of the WBC and serum levels of sTREM-1, human S100B, and proinflammatory cytokines and chemokines in the controls and severity groups, as classified by the NIHSS. While the WBC count was not significant, the serum levels of sTREAM-1, IL-6, and S-100B were significantly higher in Group 3, compared to the normal control group, Group 1, and Group 2 (Table 3). The proinflammatory cytokine, TNFα, was significantly increased in Group 3, compared with the normal controls. There was no significant difference in the other pro-inflammatory cytokines and chemokines, including IL-1β, IL-8, and IFN-γ, between the control and patient groups.

Based on the correlation analysis, the serum levels of sTREM-1 were significantly correlated with the NIHSS (*r* = 0.539, *p* < 0.001), after acute ischemic stroke (Figure 1). Furthermore, we used a regression analysis to evaluate the relationship between the serum levels of sTREM-1 and S100B. The serum levels of sTREM-1 were significantly correlated with S100B (*r* = 0.581, *p* < 0.001), after acute ischemic stroke.

### 3.4. Outcome of Stroke Is Associated with the Serum Levels of the Soluble Triggering Receptor Expressed on Myeloid Cells-1 and Human S100 Calcium-Binding Protein B and the Severity of Acute Ischemic Stroke

The Barthel index was evaluated in patients 6 months after ischemic stroke. It was shown to be significantly correlated with age, NIHSS, and the serum levels of sTREM-1 and human S100B. However, the serum levels of IL-6 and TNF-α did show a significant correlation with the Barthel index (Table 4). The results indicated that old age, a higher NIHSS, and elevated serum levels of sTREM-1 human S100B were associated with a worse stroke outcome.

### 3.5. Predictive Odds Ratios of the Serum Levels of the Soluble Triggering Receptor Expressed on Myeloid Cells-1 and Severity of Ischemic Stroke

Multiple binary logistic regression was performed to determine the risk factors of a poor outcome in patients with ischemic stroke. However, we found that S100B is significantly associated with sTREM-1 and NIHSS. Subsequently, we chose only sTREM-1 and NIHSS as the factors for the final binary logistic regression analysis. We found that the serum levels of TREM-1 and NIHSS were significantly associated with predictive factors of functional outcomes in patients with ischemic stroke (Table 5).

## 4. Discussion

The present study confirmed that the TREM-1 signaling pathway in patients with acute ischemic stroke may be activated by ischemic damaged brain tissue, increase the concentration of sTREM-1 and proinflammatory cytokines (TNFα and IL-6), and be released into circulation from ischemic stroke-damaged BBB. We found that the serum levels of sTREM-1, TNFα, IL-6, and S100B were correlated with the severity of stroke, as evaluated by TOAST and NIHSS after acute ischemic stroke. Additionally, the serum levels of sTREM-1 were significantly positively correlated with S100B. The functional outcomes were evaluated 6 months after stroke by the Barthel index, which was correlated with age and the serum levels of sTREM-1 and S100B.

In humans, TREM-1 is widely expressed on non-immune cells (microglial, epithelial cells), immune cells, and tissues, such as lymph nodes, the spinal cord, and the gastrointestinal tract [43,44]. According to recent research, TREM-1 plays an amplifier role in inflammatory processes, including synergy, with a substantial amplification of the recruitment of proinflammatory cytokines, such as TNF-α, IL-1β, IL-6, IL-8, and monocyte chemoattractant protein-1 [45,46,47]. TREM-1 has a transmembrane pattern embedded in the cell surface, with a short intracytoplasmic tail, called membrane-bound TREM-1. Because the intracytoplasmic domain does not contain well-characterized signaling motifs, it must be combined with an associated signaling unit to activate downstream intracellular pathways [20,27]. Many microorganisms or danger signals have been proposed as inducers of its activation or the secretion of sTREM-1 from the proteolytic cleavage or membrane shedding of full-length TREM-1 [27,47,48,49]. The activation of sTREM-1 has been suggested as a putative anti-inflammatory effect [30,47]. In macrophage, TREM-1 may activate phosphoinositide 3-kinases and mitogen-activated protein kinase signaling pathways to promote the integrity of mitochondria and macrophage survival through the inactivation of pro-apoptotic factors and the inhibition of the cytochrome *c* release from mitochondria [50]. Moreover, in patients with subarachnoid hemorrhage [32,33] or Alzheimer’s disease [35], the serum or cerebrospinal fluid (CSF) concentrations of sTREM-1 have a positive correlation with disease progression, severity, and prognosis. In experimental ischemic stroke [29] and subarachnoid hemorrhage [17], microglial TREM-1 expression was upregulated, following cerebral ischemic or hemorrhagic injury, which exacerbates neuroinflammatory damage. Therefore, evidence has shown a diverse contribution of the TREM-1 receptor in these scenarios, with beneficial or detrimental activities, depending on the disease and experimental model considered [17,29,35,47,48,50,51,52].

In the present study, we noted that the level of circulatory sTREM-1 was significantly correlated with severity, including infarct volume and NIHSS, and functional outcomes (Barthel index) in our patients, after acute ischemic stroke. We suggested that the acute ischemic insult is a danger signal which activates sTREM-1 from the shedding of membrane TREM-1. The sTREM-1 substantial amplifies the neuroinflammatory response and increases the release of proinflammatory cytokines from ischemic damaged tissues. Therefore, proinflammatory cytokines, including TNFα, IL-6, and sTREM-1, leaked into peripheral circulation from CSF through disrupted BBB. Proinflammatory cytokines in the acute-phase response, particularly TNFα and IL-6, were found to be associated with early clinical neurological deterioration and long-term outcomes, after acute ischemic stroke [11,19,53,54,55]. In the present study, the higher serum level of sTREM-1 may be regarded as a serum biomarker that reflects the severity of innate inflammation and poor prognosis in patients, after acute ischemic stroke. In an animal study, Liu et al. [30] suggested that within hours after ischemic stroke, TREM-1, driven by peripheral TREM1+ myeloid cells, was induced peripherally in CD11b^+^CD45^+^ cells trafficking to the ischemic brain. This may magnify the detrimental component of the post-stroke innate immune response and result in the exacerbation of ischemic brain damage [30,31]. Therefore, the post-stroke immune response offers an extended window for therapeutic intervention in acute ischemic stroke by targeting TREM-1 pathways [29,30,56]. However, whether the peripheral TREM-1 amplifies the stroke severity in patients after acute ischemic stroke is unknown. Further human studies are needed to verify this concept.

S100B is a calcium-binding protein produced mainly by astrocytes in the central nervous system, which has been implicated in the development and maintenance of the nervous system [57]. The S100B level in the blood has been used as a marker for the extent of acute damage to the brain parenchyma and BBB disruption [57,58]. Evidence has shown that the serum level of S100B was significantly correlated with the infarct volume and the levels were higher in acute stroke patients at risk of malignant infarction or hemorrhagic transformation [59,60,61]. S100B appears to be highly correlated with the extent of acute damage to the brain parenchyma, and its correlation with sTREM-1 levels may actually reflect the correlation between infarct size and neuroinflammatory response. In addition, the serum levels of S100B obtained 48 and 72 h after acute stroke provided the highest predictive values with respect to functional outcomes and infarct volume [60]. We found that the serum level of S100B was significantly correlated with severity, including the infarct volume and NIHSS in our patients with acute ischemic stroke. Apart from age being a confounding factor [62], we noted that the serum levels of S100B and sTREM-1 were highly correlated with functional outcomes, evaluated by the Barthel index 6 months after acute ischemic stroke. Higher serum levels of sTREM-1 and S100B were associated with the severity of ischemic brain damage and predisposed patients to neurological deterioration and poor functional outcomes. Furthermore, we have shown that the serum levels of sTREM-1 have a positive correlation with S100B (*r* = 0.581, *p* < 0.001). An elevated serum sTREM-1 may indicate advanced ischemic brain damage and a severe disruption of BBB, which is reflected in increased serum S100B levels in patients, after acute ischemic stroke. Thus, we suggested that the serum levels of sTREM-1 may be used as a surrogate marker for infarct severity and to predict long-term outcomes in patients after acute ischemic stroke. However, further clinical studies with a larger sample size are needed to confirm this concept.

Based on the findings presented in this study, we proposed that acute ischemic stroke may cause ischemic brain tissue damage and disrupt BBB. The ischemic brain damage will activate soluble sTREM-1 from the shedding of membrane TREM-1 in activated glial cells that promote the increased production of proinflammatory cytokines/chemokines. Figure 2 illustrates our findings and the mechanism proposed in the present study.

## 5. Conclusions

Neuroinflammation is a key component in the ischemic cascade that results in cell damage and death after cerebral ischemia. In the present study, we demonstrated that the serum levels of sTREM-1, S100B, and proinflammatory cytokines are correlated with infarct severity and long-term outcomes in patients, after acute ischemic stroke. Thus, we suggest that acute ischemic stroke induces neuroinflammation by the activation of the TREM-1 signaling pathway and the downstream inflammatory machinery that modulates the inflammatory response and ischemic neuronal cell death. The serum levels of sTREM-1 may be used as a surrogate marker in patients for the evaluation of stroke severity and functional outcomes after acute ischemic stroke. From a translational perspective, our results may allow for the development of a new therapeutic strategy for acute ischemic stroke by targeting the TREM-1 signaling pathway.

## Figures and Tables

**Figure 1 jcm-10-00061-f001:**
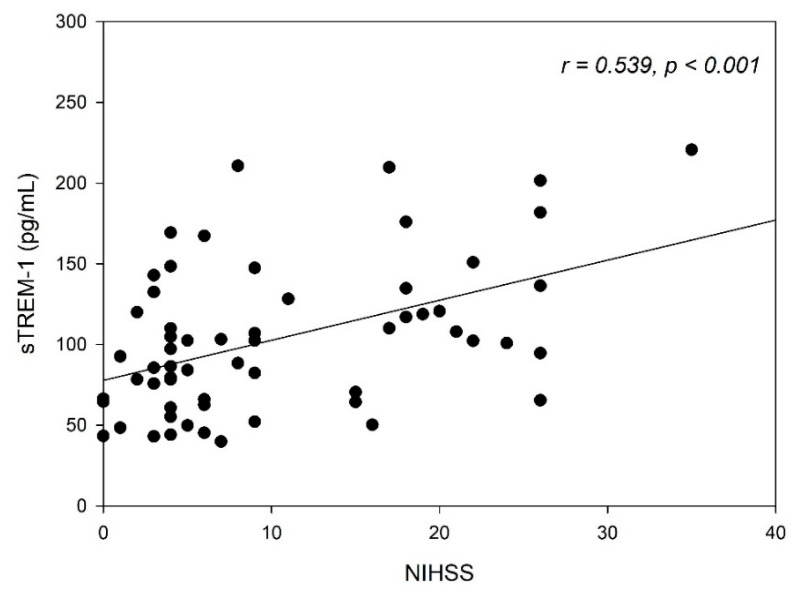
Serum levels of the soluble triggering receptor expressed on myeloid cells-1 (sTREM-1) in acute ischemic stroke patients, which are significantly correlated with the National Institutes of Health Stroke Scale (NIHSS).

**Figure 2 jcm-10-00061-f002:**
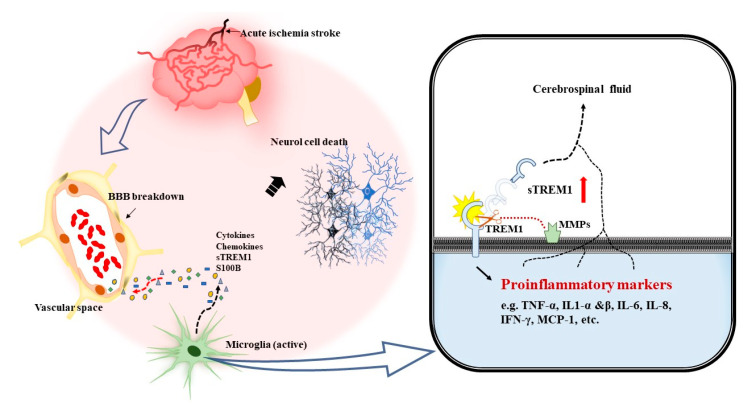
Schematic diagram illustrating the proposed triggering receptor expressed on the myeloid cells-1 (TREM-1) signaling pathway involved in downstream inflammatory machinery, which contributes to acute ischemic neuronal cell death. Acute ischemic stroke may cause ischemic brain cell death, which is influenced by the amount of human S100 calcium-binding protein B (S100B). Acute ischemic insult is a danger signal, which activates soluble TREM-1 (sTREM-1) from the shedding of membrane TREM-1 in activated glial cells, promoting an increased production of proinflammatory cytokines/chemokines in cerebrospinal fluid and leakage to serum from disrupted BBB. TNF-α: tumor necrosis α, IL-1α: interleukin 1α, IL-1β: interleukin 1β IL-6: interleukin 6, IL-8: interleukin 8, IFN-γ: interferon-γ, MCP-1: monocyte chemoattractant protein-1, MMPs, matrix metalloproteinases.

**Table 1 jcm-10-00061-t001:** Demographic data of 60 patients with stroke.

	Small Infarction	Large Infarction	
	Small-Artery Occlusion (Lacune)	Large-Artery Atherosclerosis	Cardioembolism
Sex (female/male; %)	11/22; 50%	6/10; 60%	3/8; 37.5%
Age	66.48 ± 9.81	66.25 ± 11.69	70.90 ± 10.03
Hypertension (*n*)	31	13	10
Diabetes (*n*)	9	5	6
Hyperlipidemia (*n*)	21	11	2
Ischemic heart disease (*n*)	9	4	9
Atrial fibrillation (*n*)	0	0	11
Smoking (*n*)	11	5	3

Values are expressed as the mean ± standard deviation.

**Table 2 jcm-10-00061-t002:** Comparisons of WBC, sTREM-1, proinflammatory cytokines, and S100B in terms of the infarct size in acute ischemic stroke.

	Controls (*n* = 24)	Small Infarction (*n* = 33)	Large Infarction (*n* = 27)
WBC (10^6^/mL)	6.28 ± 1.64	7.32 ± 2.32	7.87 ±1.84
sTREM-1 (pg/mL)	65.25 ± 20.96	89.20 ± 35.61	120.61± 52.39 *^,‡^
IL-1β (pg/mL)	0.36 ± 0.13	0.28 ± 0.07	0.32 ± 0.16
IL-6 (pg/mL)	0.05 ± 0.04	0.06 ± 0.04	0.56 ± 1.21 *
IL-8 (pg/mL)	4.53 ± 2.64	3.69 ± 4.10	7.53 ± 6.52
TNFα (pg/mL)	5.81 ± 5.66	6.77 ± 2.57	9.44 ± 6.83 *
IFN-γ (pg/mL)	0.63 ± 0.56	0.40 ± 0.20	0.41 ± 0.20
S100B (pg/mL)	29.96 ± 6.86	30.13 ± 15.01	59.87 ± 32.27 *^,‡^

Values are expressed as the mean ± standard deviation. WBC: white blood cell count; sTREM-1: soluble triggering receptor expressed on myeloid cells-1; IL-1β: interleukin 1β; IL-6: interleukin 6; IL-8: interleukin 8; TNF-α: tumor necrosis factor α; IFN-γ: interferon-γ; S100B: human S100 calcium-binding protein B. The Kruskal–Wallis test was used, as appropriate, to compare the continuous variables between groups. Mann–Whitney U tests were applied to compare the difference between the two groups. * *p* < 0.05 versus the control group, and **^‡^**
*p* < 0.05 versus the small infarction group.

**Table 3 jcm-10-00061-t003:** Comparisons of WBC, sTREM-1, proinflammatory cytokines, and S100B in terms of the NIHSS in acute ischemic stroke.

	Controls (*n* = 24)	Group 1 (NIHSS ≤ 5) (*n* = 25)	Group 2 (NIHSS: 6–16) (*n* = 18)	Group 3 (NIHSS ≥ 17) (*n* = 17)
WBC (10^6^/mL)	6.28 ± 1.64	6.98 ± 2.27	7.32 ± 1.59	8.25 ± 2.17
sTREM-1 (pg/mL)	65.25 ± 20.96	88.78 ± 33.94	90.64 ± 47.25	138.18 ± 44.83 *^,‡,#^
IL-1β (pg/mL)	0.36 ± 0.13	0.32 ± 0.14	0.26 ± 0.007	0.33 ± 0.17
IL-6 (pg/ml)	0.05 ± 0.04	0.07 ± 0.04	0.10 ± 0.08	1.80 ± 3.09 *^,‡,#^
IL-8 (pg/mL)	4.53 ± 2.64	5.76 ± 7.89	4.18 ± 2.20	8.43 ± 3.17
TNFα (pg/mL)	5.81 ± 2.66	7.27 ± 2.95	7.68 ± 7.33	11.35 ± 6.89 *
IFN-γ (pg/mL)	0.63 ± 0.56	0.40 ± 0.20	0.40 ± 0.16	0.43 ± 3.17
S100B (pg/mL)	29.96 ± 6.86	31.04 ± 15.24	45.52 ± 19.52	89.18 ± 6.86 *^,‡,#^

Values are expressed as the mean ± standard deviation. NIHSS: National Institutes of Health Stroke Scale; WBC: white blood cell count; sTREM-1: soluble triggering receptor expressed on myeloid cells-1; IL-1β: interleukin 1 beta; IL-6: interleukin 6; IL-8: interleukin 8; TNF-α: tumor necrosis factor α; IFN-γ: interferon-γ; S100B: human S100 calcium-binding protein B. The Kruskal–Wallis test was used, as appropriate, to compare the continuous variables between groups. Mann–Whitney U tests were applied to compare the difference between the two groups. * *p* < 0.05 versus the control group, **^‡^**
*p* < 0.05 versus Group 1, and ^#^
*p* < 0.05 versus Group 2.

**Table 4 jcm-10-00061-t004:** Correlation between the Barthel index and age, sTREM, IL-6, TNFα, and S100B, and NIHSS.

	Barthel Index	
	Correlation	*p* Value
Age (years)	−0.231	0.246
sTREM-1 (pg/mL)	−0.525	0.005 *
IL-6 (pg/mL)	−0.362	0.106
TNFα (pg/mL)	−0.043	0.848
S100B (pg/mL) NIHSS	−0.574 −0.686	<0.002 * <0.001 *

NIHSS: National Institutes of Health Stroke Scale; sTREM-1: soluble triggering receptor expressed on myeloid cells-1; IL-6: interleukin 6; TNF-α: tumor necrosis factor α; S100B: human S100 calcium-binding protein B. Spearman rank correlation was used to explore the relationship between the Barthel index and age, the serum levels of sTREM-1, IL-6, TNFα, and S100B, and NIHSS. * *p* < 0.05.

**Table 5 jcm-10-00061-t005:** Binary logistic regression models for risk factors and the Barthel index in patients with stroke.

	Odds Ratio	95% Confidence Interval	*p* Value
sTREM-1	1.327	1.020–1.726	0.035 *
NIHSS	1.043	1.001–1.087	0.042 *
**Constant**	0.001		0.205

Multiple binary logistic regression was used to determine the serum levels of the soluble triggering receptor expressed on myeloid cells-1 (sTREM-1) and National Institutes of Health Stroke Scale (NIHSS) associated with the Barthel index in patients with ischemic stroke. Multiple binary logistic regression was performed using the forward stepwise method. * *p* < 0.05.

## Data Availability

The datasets generated during and/or analyzed during the current study are available from the corresponding author (Yao-Chung Chuang) on reasonable request.

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
