# Peer review of "Serum Levels of Soluble Triggering Receptor Expressed on Myeloid Cells-1 Associated with the Severity and Outcome of Acute Ischemic Stroke"

_jcm, 2020, doi:10.3390/jcm10010061_

Round 1

Reviewer 1 Report

the paper by Huang et al., reports on the correlation between serum levels of soluble TREM1 and inflammatory markers and clinical measures of stroke severity, in particular underscoring the elevation of TREM1 in large infacts and in patients with poor prognosis.

The core findings are in agreement with previous studies but, considering that to date we know very little of trem1 in stroke, still provide enough information to be considered worth of publication. The manuscript is nevertheless poorly written with many typos, sentences difficult to understand or cut short; moreover, some tables  titles and figure legends contain mistakes (e.g., the word “correlation” is used instead of “comparison”). A throughout revision of the manuscript is needed.

In addition, I would reconsider the interpretation of S100B as a marker of blood-brain-barrier: s100B appears to be highly correlated with the extent of acute damage to the brain parenchima, and its correlation with trem1 levels may actually reflect the correlation between infarct size and inflammatory response.  The authors should either measure soluble claudin-5 levels or tone down their claims about what S100B actually means.

The statistics employed are also not clear: a multivariate regression analysis that takes into consideration demographic and clinical confounders should be included, but it is not clear if the authors used this  approach. Also, considering the inclusion of cardioembolic and non-embolic strokes, the pathogenic mechanism should be also considered in analysis.

Author Response

Response to Review #1

We are very appreciate your comment.

  1. The core findings are in agreement with previous studies but, considering that to date we know very little of trem1 in stroke, still provide enough information to be considered worth of publication. The manuscript is nevertheless poorly written with many typos, sentences difficult to understand or cut short; moreover, some tables  titles and figure legends contain mistakes (e.g., the word “correlation” is used instead of “comparison”). A throughout revision of the manuscript is needed.

Response: thank you for your suggestion. The English have been sent to MDPI English editing # English Editing ID english-25062. Statistical analysis of this work have been revised by our Biostatistics Center, Kaohsiung Chang Gung Memorial Hospital.

  1. In addition, I would reconsider the interpretation of S100B as a marker of blood-brain-barrier: s100B appears to be highly correlated with the extent of acute damage to the brain parenchima, and its correlation with trem1 levels may actually reflect the correlation between infarct size and inflammatory response.  The authors should either measure soluble claudin-5 levels or tone down their claims about what S100B actually means.

Response: thank you for your suggestion. We have added this notion in our Introduction: Line 92-98, discussion (Lines 328-334). However, we did not measure the soluble claudin-5 levels in this study. We are appreciated your suggestion, and we will measure the soluble claudin-5 levels in the further study.

  1. The statistics employed are also not clear: a multivariate regression analysis that takes into consideration demographic and clinical confounders should be included, but it is not clear if the authors used this approach. Also, considering the inclusion of cardioembolic and non-embolic strokes, the pathogenic mechanism should be also considered in analysis.

Response: Thank you for your suggestion. We have included a multivariate regression analysis for risk factors and Barthel index in patients with stroke by Binary logistic regression models (Methods: 2.4. Statistical Analyses; Results: 3.5. Predictive odds ratios of the serum levels of the soluble triggering receptor expressed on myeloid cells-1 and and severity of ischemic stroke and Table 5). We also revised the table 4 due to some mistake acooring the suggestion of the statistical analysis of this work have been suggested by our Biostatistics Center, Kaohsiung Chang Gung Memorial Hospital. However, the pathogenic mechanism is poor analysis may be related to small case number. Thank you for your comment, we will consider further analysis in the future studies.

Reviewer 2 Report

This paper aimed to investigate the role of the triggering receptor expressed on myeloid cells-1 (TREM-1) and its effect on the outcomes of acute ischemic stroke by using a single-center, prospective, observational study

The authors state that the role of TREM-1 as a modulator of inflammation makes it a good predictor of stroke severity and long-term outcomes. They also claim that identifying this marker could lead to novel therapeutic strategies targeting the TREM-1 pathway.

Major:

·         The English needs to be revised. There is also a lack of structure throughout. It is hard to follow at points, and parts need to be revised in order to improve the flow and to make certain claims more concise.

·        Article title and titles of sections and tables: All titles need to convey the main message of a section/table. It is currently not clear from reading titles within the article.

·         A structured abstract would be suited to this study (Objective, Methods, Results, Conclusion) as it is currently difficult to follow. 

·         There is a lack of a clear statement of hypothesis. This should be explicitly stated at the end of the introduction and then repeated in the discussion which would assist in making this more targeted.

·         Exclusion criteria:

-       While it is acknowledged that time to treatment is paramount in acute stroke, the later time points in patients who underwent hyperacute management would be useful. Any potential treatment that arises from the TREM-1 patient in order to be effective would need to be an adjunct to removing the mechanical obstruction of a clot. This population of patients may behave very differently (both from the underlying disease process and the treatment there of) and at least acknowledging this is relevant in this study and for future studies. This is also true for predicting outcomes in a large proportion of patients presenting with acute ischemic stroke.

-       There is mention of many inflammatory conditions and events (surgery or trauma), however it is not explicitly stated that patients on immunomodulatory therapy at baseline were excluded.

·         The study size is relatively small given the number of strokes in the population. Clarity of the time period over which the study was conducted should be included.

·        Discussion. Figure 2 is a very helpful addition, however it is felt that the discussion would be more focused if it contained more evidence directly related to this mechanism.

Minor:

-      In Table 1, the control group should be added for clarity (though it is acknowledged that they are mentioned in the text above).

-      Although it is acknowledged that the numbers female:male are helpful in Table 1, a decimal ratio would aid for more instant comparison

-      A Supplementary Table for the parameters for exclusion criteria would be helpful: (e.g. how was brain injury defined, what was the time cut-off for recent surgery, how was renal/hepatic insufficiency evaluated)?

Author Response

Response to Reviewer #2

We are very appreciate you suggests ti improve our maniscript

  1. The English needs to be revised. There is also a lack of structure throughout. It is hard to follow at points, and parts need to be revised in order to improve the flow and to make certain claims more concise.

Response: The English have been sent to MDPI English editing # English Editing ID english-25062.

  1. Article title and titles of sections and tables: All titles need to convey the main message of a section/table. It is currently not clear from reading titles within the article.

Response: We have clarified our subtitle in sections, tables, and Figure. Also, the statistical analysis of this work have been suggested by our Biostatistics Center, Kaohsiung Chang Gung Memorial Hospital. The English have been sent to MDPI English editing # English Editing ID english-25062.

  1. A structured abstract would be suited to this study (Objective, Methods, Results, Conclusion) as it is currently difficult to follow.

Response: the style of abstract is according to the Journal’s guideline.

  1. There is a lack of a clear statement of hypothesis. This should be explicitly stated at the end of the introduction and then repeated in the discussion which would assist in making this more targeted.

Response: Thank you for your suggestion. We have added the hypothesis in Lines 92-94.

  1. There is mention of many inflammatory conditions and events (surgery or trauma), however it is not explicitly stated that patients on immunomodulatory therapy at baseline were excluded.

Response: Thank you for your suggestions. We have revised our exclusion criteria. (Lines 113-118)

  1. The study size is relatively small given the number of strokes in the population. Clarity of the time period over which the study was conducted should be included.

Response: The relatively small study number was related to the time period is short and we only selected first-even acute stroke. The time period is from October 2018 to October 2019, we have indicated in Line 108.

  1. Figure 2 is a very helpful addition, however it is felt that the discussion would be more focused if it contained more evidence directly related to this mechanism.

Response: Figure 2 illustrating the proposed TREM-1 signaling pathway involved in downstream inflammatory machinery that contribute to acute ischemic neuronal cell death. Our hypothesis has also been discussed in our “Discussion” section. We used Figure 2 to conclude our hypothesis and explained the mechanism in Legend of Figure 2. Also we move it toe discussion and add a discussion in Lines 348-352.

  1. In Table 1, the control group should be added for clarity (though it is acknowledged that they are mentioned in the text above).

Response: the control group is health, age and sex have been mentioned in the text.

  1. Although it is acknowledged that the numbers female:male are helpful in Table 1, a decimal ratio would aid for more instant comparison

Response: Thank you for your suggestion. We have added the information in Table 1.

  1. A Supplementary Table for the parameters for exclusion criteria would be helpful: (e.g. how was brain injury defined, what was the time cut-off for recent surgery, how was renal/hepatic insufficiency evaluated)?

Response: we have added a supplementary material for the exclusion criteria.